# SGLT2 Inhibitors Are Associated with Left Ventricular Reverse Remodeling in Patients with Non-Compaction Cardiomyopathy—A Prospective Observational Cohort Trial

**DOI:** 10.3390/biomedicines13112773

**Published:** 2025-11-13

**Authors:** Andraž Cerar, Gregor Poglajen, Gregor Zemljič, Sabina Frljak, Neža Žorž, Martina Jaklič, Renata Okrajšek, Miran Šebeštjen, Bojan Vrtovec

**Affiliations:** 1Advanced Heart Failure and Transplantation Centre, Department of Cardiology, University Medical Centre Ljubljana, Zaloška 7, 1000 Ljubljana, Slovenia; gregor.poglajen@kclj.si (G.P.); gregor.zemljic@kclj.si (G.Z.); sabina.frljak@kclj.si (S.F.); neza.zorz@kclj.si (N.Ž.); renata.okrajsek@kclj.si (R.O.); miran.sebestjen@kclj.si (M.Š.);; 2Medical Faculty, University of Ljubljana, Vrazov trg 2, 1000 Ljubljana, Slovenia; 3Centre for Clinical Studies, University Medical Centre Ljubljana, Zaloška 7, 1000 Ljubljana, Slovenia; martina.jaklic@kclj.si

**Keywords:** non-compaction cardiomyopathy, guideline-directed heart failure medical therapy, myocardial recovery

## Abstract

**Background/Objectives:** Sodium glucose co-transporter 2 inhibitors (SGLT2is) improve outcomes in heart failure; however, data in left ventricular non-compaction cardiomyopathy (LVNC) patients are limited. We sought to analyze the clinical effects of the SGLT2is dapagliflozin and empagliflozin in patients with LVNC. **Methods:** Thirty consecutive LVNC patients diagnosed by CMR were prospectively enrolled. Clinical, biochemical and echocardiography data were obtained at the initiation of the SGLT2is and at the 12-month follow-up. All patients were on stable guideline-directed medical therapy. A response to SGLT2i therapy was defined as an improvement in LVEF ≥ 5% at 12 months. **Results:** Of the 30 enrolled patients, 25 were male, with a mean age of 49 ± 16 years and few comorbidities. Dapagliflozin 10 mg was prescribed to 23 patients and empagliflozin 10 mg to 7 patients. Five patients experiened an adverse event during follow-up (one sudden cardiac death; four heart transplantations or LVAD implantations). During follow-up, significant improvements were observed in LVEF (32.1 ± 6.9% vs. 43.5 ± 9.7%; *p* = 0.003), LVOT VTI (14.8 ± 6.5 cm vs. 17.6 ± 3.3 cm; *p* = 0.008), E/e′ (14.8 ± 4.7 vs. 10.0 ± 4.1; *p* < 0.001), and TAPSE (2.0 ± 0.4 cm vs. 2.3 ± 0.4 cm; *p* = 0.012). NT-proBNP levels decreased significantly (2025 ± 2198 pg/mL vs. 582 ± 803 pg/mL; *p* = 0.005). Eighteen patients responded favorably to SGLT2i therapy (Group A), whereas seven showed no significant LVEF improvement (Group B). The groups did not differ significantly in age, sex, baseline creatinine, or bilirubin. Compared to Group B, Group A had a smaller baseline LV end-diastolic diameter (6.3 ± 0.8 cm vs. 7.1 ± 0.9 cm; *p* = 0.025) and lower NT-proBNP levels (1720 ± 1662 pg/mL vs. 4527 ± 4397 pg/mL; *p* = 0.02). **Conclusions:** In patients with LVNC, SGLT2i therapy is associated with significant reverse remodeling and functional improvement. Benefits may be greater in those with less advanced disease.

## 1. Introduction

Left ventricular non-compaction cardiomyopathy (LVNC) is a rare myocardial disorder characterized by a non-compacted inner layer with prominent trabeculations and deep intertrabecular recesses, overlying a compacted outer layer [1,2]. Even though the disease mainly affects the left ventricle (LV), the right ventricle (RV) can also be affected [2]. LVNC is thought to be a phenotype of cardiomyopathy, being a consequence of different genetic variants that lead to an arrest of the myocardial compaction process in the early embryonic period, resulting in persistent myocardial trabeculations [3]. Recently, however, several authors have debated that LVNC could also arise from pathological volume-loading conditions of the left ventricle. Reversible hypertrabeculation, which can be physiologically observed in athletes or during pregnancy, should be distinguished from LVNC [4].

Diagnosis relies on imaging, traditionally echocardiography and more recently cardiac magnetic resonance (CMR), using established non-compacted to compacted (NC/C) ratios. According to Jenni criteria, an NC/C ratio > 2:1 in systole on echocardiography or >2.3:1 in diastole on CMR supports LVNC diagnosis [5,6].

Recent guidelines, published by the European Society of Cardiology, classify LVNC as a phenotypical trait; however, guidelines of American College of Cardiology clearly define LVNC as a genetic cardiomyopathy [7,8]. Common genetic variants involve sarcomeric (MYH7, MYBPC3, TTN) and mitochondrial (TAZ) genes and may be associated with congenital heart disease, arrhythmias, and neuromuscular disorders [3,9].

Patients with LVNC can be asymptomatic; however, in the course of the disease, signs of congestive heart failure, arrhythmias or thromboembolic events can appear. Most patients diagnosed with LVNC will develop symptoms of ventricular failure over the years, and as many as 47% of adults (and 75% of symptomatic patients) die within 6 years of its presentation [2]. Sudden cardiac death has been described as the first clinical manifestation, this also being the case in the absence of impaired left ventricular size or function [10].

Current therapeutic options in patients with LVNC are limited. Treatment of heart failure follows heart failure with reduced ejection fraction (HFrEF) guidelines; however, heart failure (HF) therapy has not shown improvements in the left ventricle ejection fraction (LVEF) in studies on LVNC patients [11], leading to possible heart transplantation (HTx) or left ventricular assist device (LVAD) implantation [12,13]. Case reports indicate the benefits of ARNI therapy [14,15], but the effects of SGLT2 inhibitors (SGLT2is) on myocardial reverse remodeling in LVNC remain unclear.

We thus sought to evaluate the effects of SGLT2i on myocardial reverse remodeling in patients with LVNC.

## 2. Methods

### 2.1. Patient Population

We conducted a prospective open-label, non-randomized study at the Advanced Heart Failure and Transplantation Programme, Department of Cardiology, at University Medical Center Ljubljana between 1 May 2020 and 31 May 2023.

The patient inclusion criteria were as follows: an age of 18 to 70 years; diagnosis of LVNC by echocardiography, confirmed by CMR using the Petersen criterion (i.e., a non-compacted to compacted myocardial ratio greater than 2.3:1 in diastole); established left ventricular systolic dysfunction (LVEF < 40%); and stable heart failure medical management in accordance with guidelines in force at the time, for at least 6 months. The exclusion criteria consisted of proven coronary artery disease, associated congenital heart disease and pregnancy. The study protocol was approved by the National Ethics Committee of the Republic of Slovenia (No. 0120-111/2023).

We enrolled 30 consecutive LVNC patients in the study after obtaining written informed consent.

### 2.2. Patient and Public Involvement

Patients and/or the public were not involved in the design, conduction, reporting, or dissemination plans of this research.

### 2.3. Study Design

After enrollment, we performed a detailed clinical evaluation, transthoracic echocardiography and peripheral blood biochemical analysis. The patients were started on the SGLT2is dapagliflozin or empagliflozin at the discretion of the treating cardiologist (Figure 1). During the study period, patients maintained the doses of other heart failure medications that were utilized as per current ESC guidelines [11].

At the 12-month follow-up, we repeated clinical, biochemical and echocardiographic evaluation. Five patients experienced an event during follow-up: three underwent heart transplantation, one patient underwent LVAD implantation, and one patient died due to SCD. Twenty-five patients successfully completed the 12-month follow-up and were included in the final analysis.

Patients were categorized into two groups based on their response to SGLT2i therapy after 12 months. Group A (responders) was defined as patients who demonstrated an increase in left ventricular ejection fraction (LVEF) of ≥5% from the baseline, whereas Group B (non-responders) included those with an LVEF increase < 5%. Favorable response after SGLT2i initiation was defined as an increase in left ventricular ejection fraction ≥ 5% at 12-month follow-up, reflecting a clinically meaningful improvement in LV systolic function [16].

### 2.4. Echocardiography

Echocardiographic data were recorded at the baseline and at 12 months using a Vivid E95 ultrasound system (General Electric Healthcare, Wauwatosa, WI, USA) and were analyzed at the end of the study by an independent echocardiographer blinded to the treatment and study timepoints. Left ventricular end-diastolic dimension (LVEDD) and end-systolic dimension (LVESD) were measured in the parasternal long-axis view. Left ventricular end-diastolic volume (LVEDV), left ventricular end-systolic volume (LVESV), and left ventricular ejection fraction (LVEF) were estimated using the Simpson biplane method. Left ventricle outflow tract velocity time integral (LVOT VTI) was obtained by tracing the Doppler spectrum of LVOT systolic flow from the apical five- or three-chamber view. In the apical four-chamber view, E velocity (mitral inflow velocity) was measured from pulsed-wave (PW) Doppler on mitral valve during diastole. From the same view, annular pulsed wave Doppler tissue imaging was obtained on the septal and lateral side of the mitral annulus (E_m_ and E_l_); both values were averaged to yield e′. The ratio of E/e′ was then calculated. Right ventricular function was assessed by tricuspid annular plane systolic excursion (TAPSE). All echocardiographic measurements were averaged over five cycles.

### 2.5. NT-proBNP Measurements

Blood samples were collected at baseline and at the 12-month follow-up. EDTA-coated, aprotinin-containing test tubes were used. Immediately after the blood samples were obtained, samples were placed on ice (for up to 4 h) and centrifuged at 4500 rpm for 15 min at 0 °C. After centrifugation, the serum was extracted and stored separately (at −80 °C). All NT-proBNP assays were performed using a standard commercial kit (Roche Diagnostics, Rotkreuz, Switzerland) at a hospital’s central independent laboratory, with those performing them blinded to the patients’ clinical status and medication data.

### 2.6. Statistical Methods

Continuous variables were summarized as the mean ± standard deviation (SD), and categorical variables as counts and percentages. The Shapiro–Wilk test was used to examine the normality assumption of continuous variables. Variables with a normal distribution were compared using independent-samples *t*-tests, whereas non-normally distributed variables were compared using the Mann–Whitney U test. Categorical variables were analyzed using the chi-squared test or Fisher’s exact test, as appropriate. *p* values < 0.05 were considered statistically significant. All statistical analyses were performed using IBM SPSS Statistics version 25.0 (IBM Corp., Armonk, NY, USA).

## 3. Results

### 3.1. Patient Characteristics

Baseline patient characteristics are summarized in Table 1. Of the 25 patients included in our analysis, the majority were male, with a mean age of 49 years. The mean LVEF was 32.1%, the LVEDV was 216 mL, and the TAPSE was 2.0 cm, indicating predominant left ventricular involvement. The NC/C ratio, as assessed by CMR, was >2.3:1 in diastole according to the Petersen criterion, confirming the LVNC phenotype. Late gadolinium enhancement on CMR, indicating myocardial fibrosis, was present in 12 patients (48%). Baseline NT-proBNP serum levels were, as expected, elevated.

All patients had been receiving maximally tolerated doses of guideline-directed heart failure therapy for at least six months, in accordance with the ESC Heart Failure guidelines available at that time. At baseline, 24 patients were on angiotensin receptor/neprilysin inhibitors (ARNIs), with 1 on an ACEI (due to ARNI adverse effects); all patients were receiving BB and MRAs (Table 1). After the baseline visit, 19 patients received dapagliflozin 10 mg and 6 patients empagliflozin 10 mg. Of note is that the five patients who experienced adverse clinical events were all on dapagliflozin; however, these events were not attributed to SGLT2i therapy. Due to its earlier approval and availability, most patients received dapagliflozin. There were no significant differences in the baseline characteristics between patients treated with dapagliflozin and those treated with empagliflozin. No adverse events attributable to SGLT2i therapy were reported among the study participants.

### 3.2. Effect of SGLT2i on NYHA Class

In our patient population, we observed an improvement in New York Heart Association (NYHA) functional class. At the baseline, 12 patients (48%) were in NYHA class II and 13 patients (52%) were in NYHA class III. At the 12-month follow-up after SGLT2i initiation, LVNC patients’ functional capacity improved; 17 patients (68%) were in NYHA class I and 8 patients (32%) in NYHA class II, *p* < 0.001 (Figure 2).

### 3.3. Effects of SGLT2i on Reverse Remodeling and Neurohormonal Modulation

At the 12-month follow-up, we observed a significant improvement in myocardial structure and function. We observed a significant reduction in LVESV from 149.5 ± 54.2 mL at the baseline to 111.6 ± 55.0 mL at the follow-up (Δ = −37.9 mL, 95% CI −64.0 to −11.8, *p* = 0.007). Left ventricular ejection fraction (LVEF) increased from 32.1 ± 6.9% to 43.5 ± 9.7% (Δ = +11.4%, 95% CI 7.5 to 15.3, *p* = 0.003) (Figure 3). We also observed an increase in stroke volume, as reflected by LVOT VTI (14.8 ± 6.5 cm vs. 17.6 ± 3.3 cm; Δ = +2.8 cm, 95% CI 0.95 to 4.65, *p* = 0.008), and a significant reduction in LV filling pressure (E/e′ 14.8 ± 4.7 vs. 10.0 ± 4.1; Δ = −4.8, 95% CI −7.9 to −1.7, *p* < 0.001) (Figure 3).

Furthermore, we also demonstrated a significant improvement in right ventricular function after the introduction of SGLT2is (TAPSE 2.0 ± 0.4 cm vs. 2.3 ± 0.4 cm; Δ = +0.3 cm, 95% CI 0.0 to 0.6, *p* = 0.012) (Figure 3). Our data also confirmed a beneficial effect of SGLT2is on neurohormonal activation as NT-proBNP serum levels decreased significantly at the 12-month follow-up (2026 ± 2198 pg/mL vs. 582 ± 803 pg/mL; Δ = −1444 pg/mL, 95% CI −2118 to −770, *p* = 0.005).

When comparing responders (Group A, *n* = 18) to non-responders (Group B, *n* = 7) to SGLT2i therapy (Table 1) the two groups differed significantly in LVEDD (6.3 ± 0.8 cm in Group A vs. 7.1 ± 0.9 cm in Group B, *p* = 0.025), serum sodium levels (140.6 ± 2.6 mmol/L vs. 138.2 ± 2.0 mmol/L, *p* = 0.011) and γGT levels (0.7 ± 0.5 μkat/L vs. 1.4 ± 1.1 μkat/L, *p* = 0.017). Patients in Group A also had significantly lower baseline NT-proBNP serum levels (1720 ± 1662 pg/mL in Group A vs. 4527 ± 4397 pg/mL in Group B, *p* = 0.02).

## 4. Discussion

Our analysis suggests that the addition of SGLT2is to heart failure therapy in patients with LVNC may promote structural and functional biventricular reverse remodeling. Our data further demonstrated that SGLT2i in this patient cohort leads to improvements in NYHA functional class. Importantly, our data suggest that the improvement is more pronounced in LVNC patients with less advanced forms of the disease.

To date, no specific therapy for LVNC has been described. Patients with symptomatic HFrEF, regardless of LVNC phenotype, are treated according to current guidelines, with the introduction of ARNI, ACEi or ARB, beta blockers, MRA and SGLT2i [11]. So far, only a small number of studies have evaluated the effects of HF GDMT in this patient cohort, showing some clinical benefit. Li et al. have studied the effect of beta-adrenergic blockers on possible reverse remodeling in 20 patients with LVNC with a median LVEF of 32%; however, only left ventricular mass reduction was shown in 13 ± 6 months of follow-up, with no significant change in either left ventricle end-diastolic diameter or ejection fraction. Furthermore, NT-proBNP serum levels have been measured only at the baseline with, no additional measurements upon follow-up to support the possibility of reverse remodeling [17].

The use of ARNI therapy (sacubitril/valsartan) has only been reported in a limited number of LVNC patients [14,15]. These studies suggest potential clinical benefits, including improved symptoms and modest reverse remodeling, but these studies contained small samples, were mostly retrospective, and lacked long-term follow-up. In our cohort, patients were already on optimized HF therapy, including ARNIs, beta blockers, and MRA, prior to SGLT2i initiation. This underscores the novelty of our findings, as we demonstrate additional structural and functional benefits with the introduction of SGLT2is in LVNC patients already receiving guideline-directed therapy.

In the contemporary treatment of HF, apart from the inhibition of the renin–angiotensin–aldosterone axis as a mode of action of other major heart failure drug classes, SGLT2is have consistently shown cardiovascular benefits, likely through various mechanisms, including direct metabolic effects on the myocardium, vessels and kidneys [18]. Several large clinical trials have demonstrated the clinical efficacy of SGLT2is, with reductions in heart failure-associated hospital admissions being shown, as well as reductions in cardiovascular and all-cause mortality [19,20]. A recent meta-analysis has shown that the addition of SGLT2is in HFrEF patients leads to significant reverse remodeling, namely a reduction in LVESV by 12.3 ± 3.8 mL and an improvement in LVEF by 2.8 ± 6.9% [21]. Our results are in line with these data, as we have also established comparable improvements both in LVESV reduction and LVEF improvement in LVNC patients taking SGLT2is.

The mechanisms through which SGLT2is may promote structural and functional reverse remodeling in LVNC likely extend beyond their hemodynamic effects. Recent evidence indicates that SGLT2is exert a range of pleiotropic effects (including metabolic, anti-inflammatory, and anti-fibrotic effects) that may contribute to myocardial structural and functional reverse remodeling and limit disease progression [22]. From a metabolic standpoint, SGLT2is shift myocardial substrate utilization toward ketone bodies, enhancing energy efficiency in the failing myocardium. They also attenuate myocardial inflammation by reducing oxidative stress and NF-κB-mediated cytokine signaling, and exert anti-fibrotic effects by modulating cardiac fibroblast activation and extracellular matrix deposition [23]. These insights may explain the structural and functional improvements observed in our LVNC cohort, extending the benefits of SGLT2i to this specific cardiomyopathy phenotype.

Both DAPA-HF and EMPEROR-reduced clinical trials have shown that the condition of patients with HFrEF improved by at least one NYHA functional class after SGLT2i therapy initiation, compared to a placebo [19,20]. In accordance with these data, our results have also shown improvements in NYHA functional class in LVNC patients on SGLT2i therapy at the 12-month follow-up with, with an average improvement of one NYHA class.

Alcidi et al. performed a retrospective study confirming the reverse remodeling of LV in 78 HFrEF patients after SGLT2i initiation. Additionally, an improvement in RV function was observed at the 12-month follow-up. However, they found no significant change in estimated filling pressures, as calculated by E/e′ [24]. In our study, we obtained similar findings regarding RV function improvement; furthermore, we also observed significant decreases in E/e′, which led us to the conclusion of there also having been a possible improvement in diastolic function in the LVNC patients.

The effect of SGLT2is on neurohormonal activation was also analyzed by Chen et al. in a recent meta-analysis of 18 trials with HF patients and SGLT2is; the study showed a significant reduction in NTproBNP by at least 20% (*p* = 0.02) [25]. In our study, we confirmed significant reductions in NTproBNP levels, thus also confirming the extended benefits of SGLT2is in LVNC patients.

Considering the response to SGLT2i therapy, our data suggest that patients caught earlier in the course of their disease, with smaller LV sizes, better hepatic function and less pronounced neurohormonal activation may benefit the most from SGLT2i therapy. Alternatively, patients with larger ventricles and more pronounced neurohormonal activation reflect an advanced stage of the disease, with a lower capacity for significant myocardial reverse remodeling [19].

Our study has several limitations that should be acknowledged. This study was a non-randomized, open label study; this study design, however, could be justified by its adherence to the current HF guidelines, which recommend the introduction of SGLT2is as a class IA indication for HF patients, making a randomized, placebo-controlled study ethically questionable.

Although both of the SGLT2 inhibitors used, dapagliflozin and empagliflozin, which are currently indicated for HF treatment, are comparable, dapagliflozin was more frequently introduced into therapy for our patients. This can be attributed to the earlier indication approval of dapagliflozin therapy for HFrEF by the European Medicines Agency in our country.

The exclusion of five patients who experienced an event may have introduced survivor bias, potentially leading to an overestimation of treatment efficacy. However, due to the incomplete follow-up data in these cases, a modified intention-to-treat analysis could not be reliably performed, which represents a limitation of the study.

Lastly, this study was performed on a relatively small study sample, which limits statistical power and prevents definitive conclusions. While our findings provide valuable preliminary insights into the effects of SGLT2i in LVNC, they should be interpreted with caution, as they may partly reflect the ongoing effect of background heart failure therapy or the natural progression of disease. Larger, multicenter studies that include the assessment of genetic background are needed to confirm our observations.

## 5. Conclusions

In patients with LVNC and HFrEF, the addition of SGLT2 inhibitor therapy to concomitant heart failure treatment appears to be associated with favorable clinical outcomes and may contribute to reverse remodeling of the failing myocardium in this patient population.

## Figures and Tables

**Figure 1 biomedicines-13-02773-f001:**
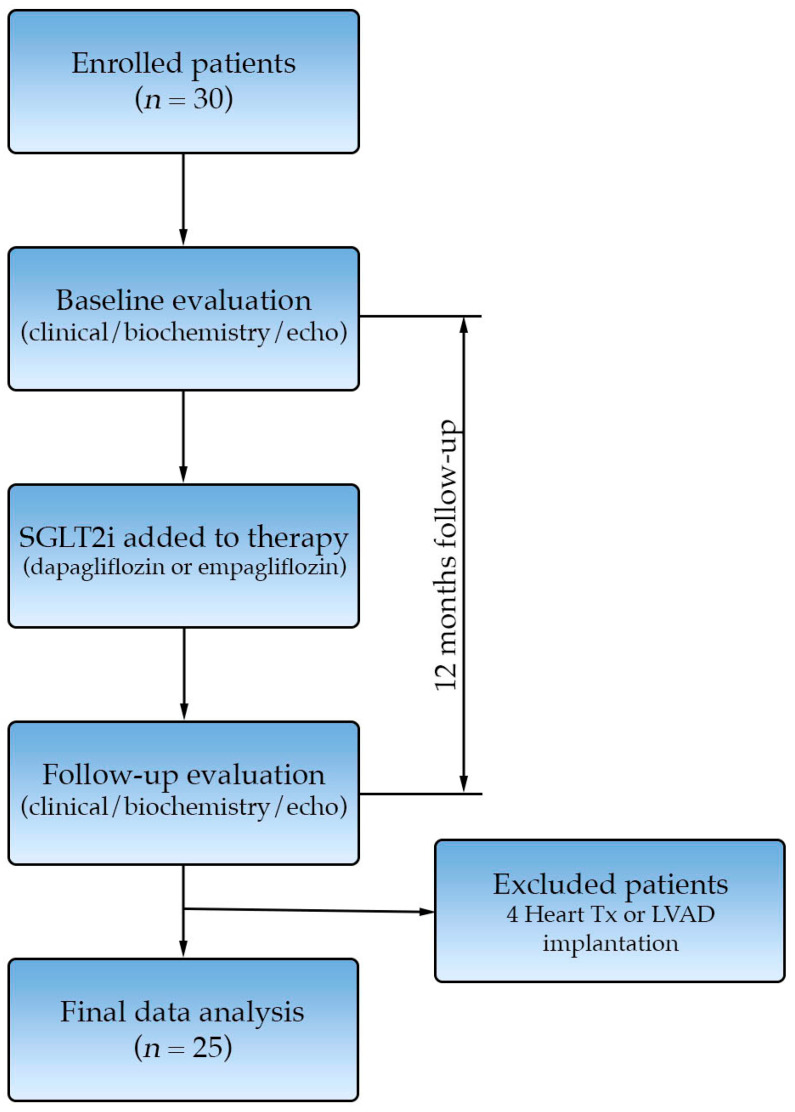
Study consort diagram.

**Figure 2 biomedicines-13-02773-f002:**
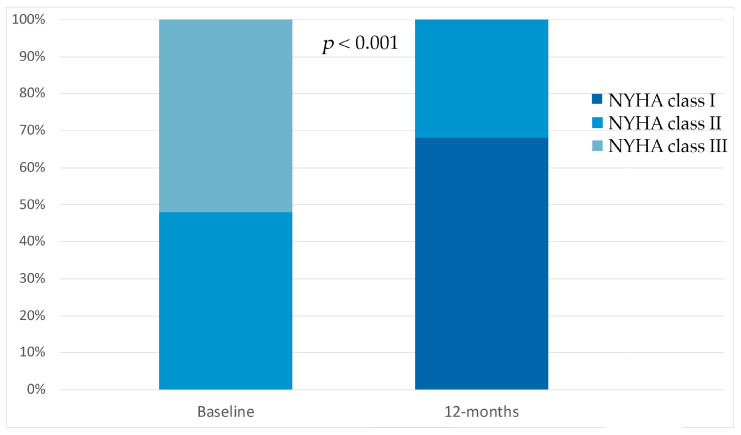
Compared to the baseline, sodium glucose co-transporter 2 inhibitor therapy showed significant improvements in New York Heart Association class at 12-month follow-up.

**Figure 3 biomedicines-13-02773-f003:**
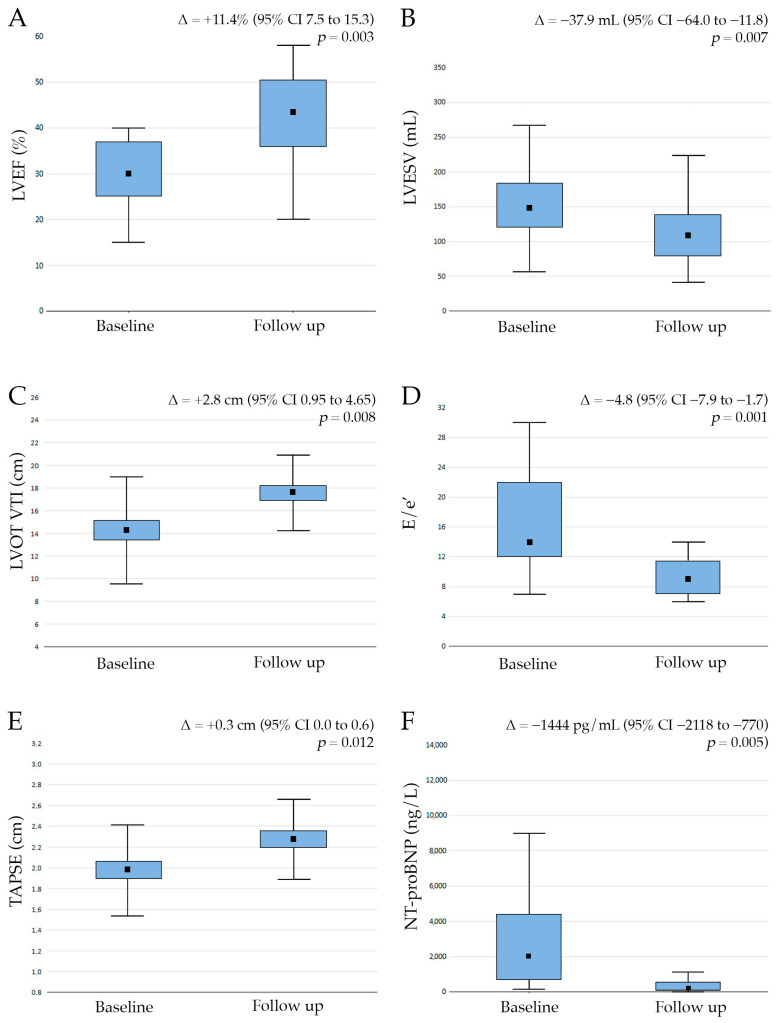
Compared to the baseline, sodium glucose co-transporter 2 inhibitor therapy showed significant improvements in left ventricular ejection fraction (LVEF, (**A**)), a decrease in left ventricular end-systolic volume (LVESV, (**B**)), an increase in left ventricle outflow tract velocity time integral (LVOT VTI, (**C**)), tricuspid annular plane systolic excursion (TAPSE, (**D**)), a decrease in the ratio of E/e′ (the E wave of the transmitral flow velocity to peak mitral annular pulsed wave Doppler tissue imaging, (**E**)) and a decrease in N-terminal pro-B-type natriuretic peptide (NT-proBNP, (**F**)) at the 12-month follow-up.

**Table 1 biomedicines-13-02773-t001:** Baseline characteristics of all patients and according to LV remodeling after SGLT2i therapy (Group A: responders, ΔLVEF ≥ 5%; Group B: non-responders, ΔLVEF < 5%).

	All	Group A (*n* = 18)	Group B (*n* = 7)	*p*
Age (years)	49 ± 16	50 ± 15	46 ± 15	0.282
Male gender	17 (68%)	12 (67%)	5 (71%)	0.538
NYHA class				
II	12 (48%)	9 (50%)	3 (43%)	0.645
III	13 (52%)	9 (50%)	4 (67%)	
Medical history				
Arterial hypertension	2 (8%)	2 (11%)	0	0.352
Diabetes	0	0	0	/
Atrial fibrillation	6 (24%)	2 (11%)	4 (57%)	0.135
ICD	12 (48%)	6 (33%)	6 (86%)	0.097
CRT	5 (20%)	3 (17%)	2 (29%)	0.696
Laboratory values				
Sodium (mmol/L)	139 ± 2.7	140.6 ± 2.6	138.2 ± 2.0	0.011
Potassium (mmol/L)	4.4 ± 0.5	4.5 ± 0.5	4.3 ± 0.4	0.352
Creatinine (μmol/L)	86 ± 25	83 ± 27	89 ± 21	0.527
Bilirubine (μmol/L)	19 ± 14	21 ± 15	17 ± 12	0.558
γGT (μkat/L)	1.0 ± 0.8	0.7 ± 0.5	1.4 ± 1.1	0.017
AST (μkat/L)	0.7 ± 0.7	0.5 ± 0.4	0.9 ± 0.9	0.208
ALT (μkat/L)	0.9 ± 1.1	0.5 ± 0.3	1.4 ± 1.7	0.052
hs-TnI (pg/mL)	13.8 ± 11.7	11.4 ± 5.7	19.9 ± 20.0	0.31
NT-proBNP (pg/mL)	2026 ± 2198	1720 ± 1661	4527 ± 4369	0.020
Medical therapy				
ACEI/ARB	1 (4%)	0	1 (14%)	/
ramipril 1.25 mg qid	1 (4%)	0	1 (14%)	/
ARNI	24 (96%)	18 (100%)	6 (86%)	/
sac/val 24/26 mg bid	2 (8%)	0	2 (29%)	/
sac/val 49/51 mg bid	11 (44%)	9 (50%)	2 (29%)	/
sac/val 103/97 mg bid	11 (44%)	9 (50%)	2 (29%)	/
Beta blockers	25 (100%)	18 (100%)	7 (100%)	/
bisoprolol 2.5 mg qid	5 (20%)	3 (17%)	2 (29%)	/
bisoprolol 5 mg qid	12 (48%)	8 (44%)	4 (57%)	/
bisoprolol 10 mg qid	5 (20%)	5 (28%)	0	/
carvedilol 6.25 mg bid	1 (4%)	1 (6%)	0	/
carvedilol 12.5 mg bid	1 (4%)	0	1 (14%)	/
carvedilol 25 mg bid	1 (4%)	1 (6%)	0	/
MRA	25 (100%)	18 (100%)	7 (100%)	/
spironolactone 25 mg qid	16 (64%)	11 (61%)	5 (71%)	/
eplerenone 25 mg qid	9 (36%)	7 (39%)	2 (29%)	/
Aspirin	7 (28%)	6 (33%)	1 (14%)	0.336
Warfarin	9 (36%)	5 (28%)	4(57%)	0.169

Abbreviations: NYHA, New York Heart Association; ICD, implantable cardioverter defibrillator; CRT, cardiac resynchronization therapy; γGT gamma-glutamyl transferase; AST, aspartate aminotransferase; ALT, alanine aminotransferase; hs-TnI, high sensitivity troponin I; NT-proBNP, N-terminal pro-B-type natriuretic peptide; ACEI, angiotensin converting enzyme inhibitor; ARB, angiotensin receptor blocker; ARNI, angiotensin receptor neprilysin inhibitor; sac/val, sacubitril/valsartan; MRA, mineralocorticoid receptor antagonist.

## Data Availability

The data presented in this study are available on request from the corresponding author.

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
