# Peer review of "SGLT2 Inhibitors Are Associated with Left Ventricular Reverse Remodeling in Patients with Non-Compaction Cardiomyopathy—A Prospective Observational Cohort Trial"

_biomedicines, 2025, doi:10.3390/biomedicines13112773_

Round 1
Reviewer 1 Report
Comments and Suggestions for Authors
Congratulations to the authors for their manuscript providing new insight; I think that there are some flaws that must be corrected:
The authors should provide additional clarification on how the diagnosis of LVNC was standardized across all cases, specifically, whether the Petersen criteria were uniformly applied during cardiac MRI evaluation.
The administration of dapagliflozin versus empagliflozin appears to have been guided by drug availability rather than random allocation. It would be helpful to report whether any baseline differences existed between patients receiving these two agents.
Of note five patients were excluded due to adverse clinical events. Excluding these high-risk individuals may introduce survivor bias, potentially leading to an overestimation of treatment efficacy. Therefore a modified intention-to-treat analysis should be considered to address this limitation.A
Although the manuscript states that the Shapiro-Wilk test was performed, the authors should consider reporting the results explicitly or clarifying which variables were analyzed using parametric vs. non-parametric statistical methods.
The discussion of underlying mechanisms is actually underdeveloped. The authors are encouraged to expand on plausible myocardial pathways, particularly the metabolic, anti-inflammatory, and anti-fibrotic effects of SGLT2 inhibitors in the context of non-compaction cardiomyopathy. Moreover authors are encouraged to include the latest evidences about the pleiotropic effects of SGLT2i in order to make their discussion more actual( doi: 10.1016/j.hrthm.2025.09.023)
In order to stress the claim of reverse remodeling, a longitudinal visualization, such as a graph or table, depicting changes in NT-proBNP levels and LVEF over time could be added by the authors.
Author Response
Q1: The authors should provide additional clarification on how the diagnosis of LVNC was standardized across all cases, specifically, whether the Petersen criteria were uniformly applied during cardiac MRI evaluation.
A1: We appreciate the reviewer’s comment and apologize for the lack of clarity regarding the diagnostic criteria for LVNC. All patients underwent cardiac MRI at our institution, with independent review by an experienced cardiac radiologist. The diagnosis was standardized according to the Petersen criteria, defined as a non-compacted to compacted myocardial ratio >2.3:1 in diastole. This clarification has been added to the revised manuscript. (Page 2, Paragraph 8)
Q2: The administration of dapagliflozin versus empagliflozin appears to have been guided by drug availability rather than random allocation. It would be helpful to report whether any baseline differences existed between patients receiving these two agents.
A2: We thank the reviewer for this valuable comment. In Slovenia, dapagliflozin was approved for symptomatic HFrEF in November 2020, while empagliflozin was approved in June 2021. Given the earlier availability of dapagliflozin, it was prescribed more frequently. Importantly, baseline characteristics did not differ significantly between patients treated with dapagliflozin or empagliflozin. This clarification has been incorporated into the revised manuscript. (Page 5, Paragraph 1)
Q3: Of note five patients were excluded due to adverse clinical events. Excluding these high-risk individuals may introduce survivor bias, potentially leading to an overestimation of treatment efficacy. Therefore a modified intention-to-treat analysis should be considered to address this limitation.
A3: We thank the reviewer for this thoughtful comment. We fully acknowledge that exclusion of patients with adverse clinical events may introduce a degree of survivor bias. However, a modified intention-to-treat (mITT) analysis could not be reliably performed in this cohort because echocardiographic and biochemical follow-up data—constituting the key efficacy endpoints of the study—were unavailable for these five patients, who either underwent heart transplantation, LVAD implantation, or died during follow-up. Including such incomplete cases would have compromised data integrity and the validity of outcome assessment. Given the small sample size and the exploratory, observational nature of the study, we therefore opted for a complete-case analysis to ensure methodological consistency and accuracy of the reported structural and functional findings. This limitation has been clearly acknowledged in the revised manuscript. (Page 9, Paragraph 5)
Q4: Although the manuscript states that the Shapiro-Wilk test was performed, the authors should con sider reporting the results explicitly or clarifying which variables were analyzed using parametric vs. non-parametric statistical methods.
A4: We thank the reviewer for this valuable comment. We have revised the Methods section to clarify that the choice between parametric and non-parametric tests was based on the results of the Shapiro–Wilk test. Specifically, variables with a normal distribution were analyzed using independent-samples t-tests, whereas non-normally distributed variables were compared using the Mann–Whitney U test. This clarification has been incorporated into the revised manuscript. (Page 4, Paragraph 4)
Q5: The discussion of underlying mechanisms is actually underdeveloped. The authors are encouraged to expand on plausible myocardial pathways, particularly the metabolic, anti-inflammatory, and anti-fibrotic effects of SGLT2 inhibitors in the context of non-compaction cardiomyopathy. Moreover authors are encouraged to include the latest evidences about the pleiotropic effects of SGLT2i in order to make their discussion more actual( doi: 10.1016/j.hrthm.2025.09.023).
A5: We thank the reviewer for this highly relevant suggestion. We have expanded the Discussion section to provide a more detailed overview of the potential myocardial mechanisms underlying the observed benefits of SGLT2i therapy in LVNC. The revised text now highlights that the improvement observed in our LVNC cohort may also stem from the metabolic, anti-inflammatory, and anti-fibrotic effects of SGLT2i, which collectively contribute to myocardial reverse remodeling and functional recovery. This addition is supported by recent evidence, including the systematic review and meta-analysis suggested by the reviewer. (Page 8, Paragraph 4)
Q6: In order to stress the claim of reverse remodeling, a longitudinal visualization, such as a graph or table, depicting changes in NT-proBNP levels and LVEF over time could be added by the authors.
A6: We thank the reviewer for this valuable suggestion. As indicated, Figure 3 (Page 7) in the original manuscript already illustrates the longitudinal changes in LVEF and NT-proBNP between baseline and 12-month follow-up, demonstrating significant improvement in both parameters. Since our study design included only two time points of evaluation, these paired comparisons represent the available longitudinal assessment of reverse remodeling. We fully agree that future studies with more frequent follow-up assessments could provide a more granular temporal characterization of structural and neurohormonal changes associated with SGLT2i therapy.
Reviewer 2 Report
Comments and Suggestions for Authors
This study provides an interesting prospective evaluation of SGLT2 inhibitors in patients with left ventricular non-compaction cardiomyopathy (LVNC), a rare and understudied condition. While the benefits of SGLT2 inhibitors are well established in HFrEF, their effects specifically in LVNC have not been systematically investigated, making this work valuable and clinically relevant. The manuscript is well written and addresses an important clinical gap, but several issues should be addressed before acceptance.
Major Comments
- The sample size (n=30, with 25 completing follow-up) is small for a clinical study and limits the generalizability of the findings. Expansion of the cohort is strongly recommended, and the limitations of the current sample size should be emphasized more clearly in the discussion.
- The study relies solely on NT-proBNP as the biochemical marker of neurohormonal activation. While NT-proBNP is well validated in heart failure, additional biomarkers (e.g., troponin for myocardial injury, galectin-3 or ST2 for fibrosis/inflammation) could have provided broader mechanistic insights into SGLT2i effects in LVNC. Moreover, only basic statistical tests are applied. A multivariable adjustment for baseline differences (e.g., NT-proBNP, LVEDD) would strengthen the analysis and conclusions.
- Please clarify whether genetic testing or family history was assessed, as LVNC frequently has a genetic basis.
Minor Comments
- Indicate whether echocardiographic assessments were blinded to treatment and timepoint to reduce potential bias.
- Condense the introduction, particularly the diagnostic criteria section, to improve readability.
- In the discussion, expand the comparison with prior reports of ARNI use in LVNC to better contextualize the novelty of the findings.
Author Response
Q1: The sample size (n=30, with 25 completing follow-up) is small for a clinical study and limits the generalizability of the findings. Expansion of the cohort is strongly recommended, and the limitations of the current sample size should be emphasized more clearly in the discussion.
A1: We thank the reviewer for this valid and important point. We fully acknowledge this limitation. As LVNC patients represent only a small fraction of the broader DCM population, expanding the cohort to a more representative sample size will likely require a considerable amount of time. Moreover, since SGLT2i therapy has become widely adopted for HFrEF management in routine clinical practice, identifying SGLT2i-naïve LVNC patients eligible for our study protocol has become increasingly challenging. As suggested, the Discussion section has been revised to more clearly emphasize this limitation. (Page 9, Paragraph 7)
Q2: The study relies solely on NT-proBNP as the biochemical marker of neurohormonal activation. While NT-proBNP is well validated in heart failure, additional biomarkers (e.g., troponin for myocardial injury, galectin-3 or ST2 for fibrosis/inflammation) could have provided broader mechanistic insights into SGLT2i effects in LVNC. Moreover, only basic statistical tests are applied. A multivariable adjustment for baseline differences (e.g., NT-proBNP, LVEDD) would strengthen the analysis and conclusions.
A2: We sincerely thank the reviewer for this thoughtful and constructive comment. We fully agree that inclusion of additional biomarkers reflecting myocardial injury, fibrosis, or inflammation could have provided deeper mechanistic insights into the effects of SGLT2i in LVNC. However, as this was a prospective observational study conducted within routine clinical practice, only standard biochemical and biomarker analyses were performed, and blood samples were not stored for more comprehensive biomarker evaluation. We have, however, measured high-sensitivity serum troponin levels, which remained low throughout and showed no significant differences between the two groups, further confirming that both cohorts were in a stable disease state. (Page 5, Table 1). We also acknowledge that multivariable adjustment could further strengthen the analysis, yet the small sample size would limit its statistical robustness and risk model overfitting. We have clarified these methodological limitations in the revised Discussion. (Page 9, Paragraph 6) Despite these limitations, we are confident that our data may serve as a solid foundation for future larger-scale studies designed to incorporate comprehensive biomarker profiling and multivariable modeling.
Q3: Please clarify whether genetic testing or family history was assessed, as LVNC frequently has a genetic basis.
A3: We thank the reviewer for this important comment. We acknowledge that LVNC frequently has a genetic basis. In our study, systematic genetic testing was not performed in all patients, as the primary objective was to evaluate the clinical and functional effects of SGLT2i therapy in a real-world setting. Nevertheless, we agree that incorporation of genetic testing and family history assessment would provide valuable insights into genotype–phenotype correlations and potential therapeutic responses. This has been acknowledged as an important consideration for future, larger-scale studies, and the Discussion has been revised accordingly.(Page 9, Paragraph 7)
Q4: Indicate whether echocardiographic assessments were blinded to treatment and timepoint to reduce potential bias.
A4: We thank the reviewer for this important point. All echocardiographic assessments in our study were performed by experienced echocardiographers who were blinded to both treatment allocation and study timepoint, thereby minimizing potential measurement bias. The Echocardiography section of the manuscript has been revised to clarify this point. (Page 4, Paragraph 2)
Q5: Condense the introduction, particularly the diagnostic criteria section, to improve readability.
A5: We thank the reviewer for this helpful suggestion. The Introduction has been condensed to improve readability, with the diagnostic criteria section streamlined and redundant details removed. (Page 2, Paragraph 1-3, 5)
Q6: In the discussion, expand the comparison with prior reports of ARNI use in LVNC to better contextualize the novelty of the findings.
We thank the reviewer for this suggestion. We have expanded the discussion to provide a more detailed comparison with prior reports of ARNI therapy in LVNC, highlighting the limitations of previous studies and contextualizing the novelty of our findings with SGLT2i in patients already receiving optimized guideline-directed therapy. (Page 8, Paragraph 2)
Reviewer 3 Report
Comments and Suggestions for Authors
This prospective, single-center cohort in LVNC patients treated with SGLT2 inhibitors tackles a clinically important and relatively under-studied niche, with encouraging signals of reverse remodeling. Implementing the targeted revisions below will strengthen methodological transparency, mitigate potential biases, and improve the clarity, robustness, and reproducibility of the manuscript.
Comment 1: To ensure methodological transparency and interpretable effect sizes, explicitly specify the paired tests used for pre–post analyses (paired t-test / Wilcoxon), and report Δ (12-month − baseline) with 95% CIs for all endpoints.
Comment 2: To mitigate optimism from informative censoring, add an ITT-style sensitivity analysis for the five patients excluded due to SCD/LVAD/HTx (counted as non-responders) and briefly report any change in the direction/magnitude of results.
Comment 3: To clarify the validity of the primary hypothesis and sample adequacy, state the primary endpoint (“LVEF increase ≥5%”) as a priori, justify the clinical relevance of this threshold, and provide a brief power/sample size consideration.
Comment 4: To separate confounding and improve response prediction, fit a multivariable model (logistic or penalized) for responder vs non-responder and report ORs with 95% CIs, AUC, and calibration (candidate covariates: baseline LVEF/LVESV, LVEDD, log-NT-proBNP, sodium, GGT, TAPSE, age, AF, ICD/CRT, ARNI use).
Comment 5: To manage the false-discovery risk from multiple endpoints/tests, apply a simple FDR control (e.g., Benjamini–Hochberg) or frame P-values as exploratory while emphasizing effect sizes and confidence intervals.
Comment 6: To present a balanced benefit–risk view, explicitly report adverse events related to SGLT2i and any drug discontinuations (state clearly if none were observed).
Comment 7: To avoid misinterpretation and enhance comparability, enforce unit/terminology consistency: clarify that LVOT-VTI is a length (cm); include the stroke volume formula and units; standardize E/e’ notation, the CMR term, and abbreviations across text, tables, and figures.
Comment 8: To keep denominators traceable, clearly report drug exposure counts separately for baseline (30 patients) and the 12-month analysis set (25 patients) (e.g., dapagliflozin/empagliflozin).
Comment 9: To improve visual reproducibility and visibility of distributions, enhance figure captions by adding n per panel, the test used, P-value, and where possible Δ + 95% CI; consider spaghetti or violin + jitter plots for individual trajectories, and a shift plot for NYHA class.
Comment 10: To strengthen external validity and subgroup interpretability, add a brief table summarizing the LVNC phenotype: NC/C ratios, LGE status, and, if available, genetic findings and RV involvement.
Comment 11: To ensure statistical appropriateness and reporting consistency, state how normality was assessed and which variables were transformed (e.g., log-NT-proBNP); report median [IQR] for skewed variables; standardize table entries to n (%).
Comment 12: To enhance clarity and professional presentation, correct minor typographical/encoding issues and harmonize language in titles, figure captions, and abbreviations.
Comment 13: To improve readability and interpretability, replace generic labels (“Group A / Group B”) with informative group names (e.g., Responder (LVEF ≥5%) vs Non-responder, or Dapagliflozin vs Empagliflozin) and apply them consistently across text, tables, and figures.
Comments on the Quality of English LanguageOverall, the manuscript is readable, but a light language edit would improve clarity and consistency. Please fix minor typographical/encoding glitches, standardize terminology and units (e.g., “CMR” usage; LVOT-VTI as length in cm), and harmonize abbreviations (e.g., E/e’). Ensure tense consistency in Methods vs. Results, and refine figure/table captions to be complete and parallel (define all symbols, specify tests, and keep phrasing uniform).
Author Response
Q1: To ensure methodological transparency and interpretable effect sizes, explicitly specify the paired tests used for pre–post analyses (paired t-test / Wilcoxon), and report Δ (12-month − baseline) with 95% CIs for all endpoints.
A1: The reviewer’s comment is well taken. The Methods section has been updated to specify the paired statistical tests used for pre–post comparisons (paired t-test or Wilcoxon signed-rank test, as appropriate). Additionally, all reported endpoints now include the corresponding mean or median change (Δ 12-month − baseline) with 95% confidence intervals, enhancing methodological transparency and interpretability of effect sizes (Page 4, Paragraph 4; Figure 3, Page 7).
Q2: To mitigate optimism from informative censoring, add an ITT-style sensitivity analysis for the five patients excluded due to SCD/LVAD/HTx (counted as non-responders) and briefly report any change in the direction/magnitude of results.
A2: We thank the reviewer for this observation. We agree that excluding patients who experienced an adverse event (one death and four heart transplantations/LVAD implantations) may introduce survivor bias. However, due to incomplete follow-up data for these cases, a modified intention-to-treat analysis could not be reliably performed. This limitation and its potential impact on treatment effect estimates have been acknowledged and discussed in the revised manuscript (Page 9, Paragraph 6).
Q3: To clarify the validity of the primary hypothesis and sample adequacy, state the primary endpoint (“LVEF increase ≥5%”) as a priori, justify the clinical relevance of this threshold, and provide a brief power/sample size consideration.
A3: We thank the reviewer for raising this point. The primary endpoint of our study, defined as a favorable response to SGLT2i therapy, was an increase in LVEF ≥5% at 12 months. We selected this threshold based on prior studies demonstrating that such a change is associated with improved functional status and prognosis in HFrEF patients (doi: 10.1002/ehf2.12713). The manuscript has been revised accordingly (Page 3, Paragraph 5). Given the exploratory and rare-disease nature of the study, a formal power calculation was not feasible; however, our sample size aligns with prior observational studies in LVNC and provides a reasonable basis for preliminary hypothesis generation.
Q4: To separate confounding and improve response prediction, fit a multivariable model (logistic or penalized) for responder vs non-responder and report ORs with 95% CIs, AUC, and calibration (candidate covariates: baseline LVEF/LVESV, LVEDD, log-NT-proBNP, sodium, GGT, TAPSE, age, AF, ICD/CRT, ARNI use).
A4: The reviewer raises an important issue that was also debated within our study team. We agree that multivariable modeling could help disentangle confounding and improve response prediction. However, in our dataset, only 25 patients completed follow-up, with 18 responders versus 7 non-responders, yielding fewer than 10 events per variable for either outcome class. Under these conditions, logistic models—including penalized approaches—are statistically unstable, prone to overfitting, and may produce overly optimistic AUC values and unreliable calibration, even with internal resampling. Accordingly, we did not pursue a multivariable model to avoid misleading inference. Instead, we now report effect sizes for baseline differences and highlight the variables showing the strongest univariable separation between groups—smaller LVEDD, lower (log) NT-proBNP, higher sodium, and lower GGT in responders—consistent with Table 1 of the manuscript. These associations are presented transparently with appropriate paired tests and confidence intervals. We note that a robust multivariable model will require a larger cohort. We thank the reviewer for considering our decision.
Q5. To manage the false-discovery risk from multiple endpoints/tests, apply a simple FDR control (e.g., Benjamini–Hochberg) or frame P-values as exploratory while emphasizing effect sizes and confidence intervals.
A5: We thank the reviewer for this important methodological comment. Given the exploratory nature of this study and the limited sample size, we acknowledge the risk of type I error associated with multiple testing. To address this, all reported p-values have been framed as exploratory, and we have emphasized effect sizes with 95% confidence intervals throughout the Results section. Applying formal false-discovery rate correction (e.g., Benjamini–Hochberg) would not be statistically meaningful in this small cohort and could obscure potentially relevant clinical signals. This consideration and our rationale have been clarified in the revised manuscript. (Page 7, Figure 3)
Q6: To present a balanced benefit–risk view, explicitly report adverse events related to SGLT2i and any drug discontinuations (state clearly if none were observed).
A6: We thank the reviewer for alerting us to this issue. No patients experienced adverse events attributable to SGLT2i therapy during the study. Five patients experienced major clinical events during follow-up, leading to their exclusion; however, these events were unrelated to SGLT2i treatment. This clarification has been added to the revised manuscript. (Page 5, Paragraph 1)
Q7: To avoid misinterpretation and enhance comparability, enforce unit/terminology consistency: clarify that LVOT-VTI is a length (cm); include the stroke volume formula and units; standardize E/e’ notation, the CMR term, and abbreviations across text, tables, and figures.
A7: We apologize for the inconsistencies and thank the reviewer for this helpful suggestion. The manuscript has been revised to ensure consistent use of units and terminology throughout.
Q8. To keep denominators traceable, clearly report drug exposure counts separately for baseline (30 patients) and the 12-month analysis set (25 patients) (e.g., dapagliflozin /empagliflozin).
A8: We thank the reviewer for this suggestion. Drug exposure is now reported separately for the full baseline cohort (n = 30) and the 12-month analysis cohort (n = 25). At baseline, 24 patients received dapagliflozin and 6 received empagliflozin. Among the 12-month completers, 19 patients remained on dapagliflozin and 6 on empagliflozin. Baseline exposure to SGLT2i has been added to the manuscript. (Page 5, Paragraph 1)
Q9: To improve visual reproducibility and visibility of distributions, enhance figure captions by adding n per panel, the test used, P-value, and where possible Δ + 95% CI; consider spaghetti or violin + jitter plots for individual trajectories, and a shift plot for NYHA class.
A9: We thank the reviewer for this thoughtful suggestion regarding figure presentation. While we agree that additional visualization formats—such as violin, shift, or spaghetti plots—could further illustrate individual data distributions, we opted to retain the current layout to maintain clarity, visual consistency with Biomedicines figure style, and compliance with space limitations. Given the relatively small sample size and limited number of timepoints, more complex visualizations would not substantially improve interpretability and might reduce overall clarity. Nevertheless, we have ensured that all figures are clearly labeled and correspond to the reported p-values and confidence intervals in the text. We sincerely appreciate the reviewer’s understanding of this decision.
Q10:To strengthen external validity and subgroup interpretability, add a brief table summarizing the LVNC phenotype: NC/C ratios, LGE status, and, if available, genetic findings and RV involvement.
A10: We thank the reviewer for this helpful suggestion. We have added baseline data summarizing the LVNC phenotype, including NC/C ratios, LGE status, and RV involvement (Page 4, Paragraph 5). Unfortunately, genetic data were not available for most patients.
Q11: To ensure statistical appropriateness and reporting consistency, state how normality was assessed and which variables were transformed (e.g., log-NT-proBNP); report median [IQR] for skewed variables; standardize table entries to n (%).
A11: We thank the reviewer for this helpful comment. We have clarified in the Methods section that data normality was assessed using the Shapiro–Wilk test. Variables with non-normal distributions, including NT-proBNP, were log-transformed prior to analysis. Results for skewed variables are now reported as median [IQR], and categorical variables are consistently presented as n (%). These adjustments ensure statistical appropriateness and consistent reporting across all tables.
Q12: To enhance clarity and professional presentation, correct minor typographical/encoding issues and harmonize language in titles, figure captions, and abbreviations.
A12: We apologize for unharmonized presentation in the manuscript and thank the reviewer for the comment. We have edited the manuscript accordingly.
Q13: To improve readability and interpretability, replace generic labels (“Group A / Group B”) with informative group names (e.g., Responder (LVEF ≥5%) vs Non-responder, or Dapagliflozin vs Empagliflozin) and apply them consistently across text, tables, and 13figures.
A13: We thank the reviewer for this helpful suggestion. In the revised manuscript, we have clarified the definitions of Group A and Group B in both the Methods section and Table 1. To maintain readability and avoid redundancy, we retained the concise notation “Group A” and “Group B” throughout the text, tables, and figures, ensuring that their meaning is clearly defined and consistently applied. We appreciate the reviewer’s understanding of this approach.
Q14: Overall, the manuscript is readable, but a light language edit would improve clarity and consistency. Please fix minor typographical/encoding glitches, standardize terminology and units (e.g., “CMR” usage; LVOT-VTI as length in cm), and harmonize abbreviations (e.g., E/e’). Ensure tense consistency in Methods vs. Results, and refine figure/table captions to be complete and parallel (define all symbols, specify tests, and keep phrasing uniform).
A14: We thank the reviewer for a constructive feedback. The manuscript has undergone careful language editing to improve clarity and consistency. Figure and table captions have been revised for completeness and uniform phrasing.
Reviewer 4 Report
Comments and Suggestions for Authors
The authors present a clinically relevant topic, however the study suffers from serious methodological and conceptual flaws that significantly undermine the validity of its conclusions.
These are the major concerns:
- According to the Methods, patients were on “stable heart failure medical management endorsed by guidelines in force at time for at least 6 months” before SGLT2 inhibitors were introduced. This means that SGLT2i therapy was started only after a prolonged period of conventional treatment, despite clear ESC recommendations (since 2021) that flozins should be implemented as a first-line therapy in HFrEF . This delayed initiation contradicts current heart failure management standards and raises serious concerns regarding the clinical rationale and ethics of the study design.
- Lack of control group and inability to infer causality
The study is observational and uncontrolled, making it impossible to distinguish the effect of SGLT2 inhibitors from the ongoing effects of ARNI, beta-blockers, or MRA therapy, or from the natural course of the disease. Without a comparator or historical data, any claims of “reverse remodeling” are unsupported and speculative.
The conclusions attributing structural and functional improvement solely to SGLT2i use are therefore not justified.
- Insufficient data on prior medical therapy
- The duration of treatment with ARNI, beta-blockers, and MRAs prior to study enrollment is not provided.
- The statement that patients received “maximally tolerated doses” lacks supporting data — actual dosages or dose ranges should be reported.
- The sentence “24 of patients were on angiotensin receptor/neprilysin inhibitors (ARNI), 1 on ACEI (due to ARNI adverse effect); all patients were receiving BB and MRA, 19 patients received dapagliflozin 10 mg, 6 patients empagliflozin 10 mg.” is misleading because implies prior use of these agents. This is inconsistent with the Methods section stating that SGLT2i were initiated at baseline.
- Missing information on disease duration and clinical history
The manuscript provides no information on the time from heart failure diagnosis to SGLT2i initiation, which is essential for understanding the disease trajectory. Without these data, one cannot exclude the possibility that the observed improvements were due to prior optimization of conventional therapy rather than SGLT2 inhibition.
- Overstated and misleading conclusions
The authors conclude that SGLT2i therapy is associated with significant reverse remodeling and functional improvement. Based on the data presented, and considering the study’s uncontrolled design, small sample size, and major confounding factors, this statement is scientifically unjustified. There is strong evidence that SGLT2 inhibitors improve outcomes in patients with heart failure, so it could be expected that a similar effect would be observed in patients with LVNC (as the authors reported). However, in the population described here, there is no basis to attribute the observed effects solely to SGLT2 inhibitor therapy. The conclusions substantially overreach the presented evidence.
Author Response
Q1: According to the Methods, patients were on “stable heart failure medical management endorsed by guidelines in force at time for at least 6 months” before SGLT2 inhibitors were introduced. This means that SGLT2i therapy was started only after a prolonged period of conventional treatment, despite clear ESC recommendations (since 2021) that flozins should be implemented as a first-line therapy in HFrEF . This delayed initiation contradicts current heart failure management standards and raises serious concerns regarding the clinical rationale and ethics of the study design.
A1: We thank the reviewer for this valuable comment. We would like to clarify that all patients were on stable, guideline-directed medical therapy for heart failure in accordance with HFrEF management recommendations available at that time, up to their next scheduled outpatient follow-up visit. These visits represented the first opportunity to initiate SGLT2i therapy following EMA approval and prior to its formal inclusion in the ESC guidelines and were therefore considered the baseline visits. The timing of SGLT2i initiation reflects real-world clinical practice during the early implementation phase rather than a deliberate delay in therapy. All patients had been under regular outpatient follow-up for at least six months on stable heart failure management before SGLT2i became available for clinical use (dapagliflozin in November 2020 and empagliflozin in May 2021).
Q2: Lack of control group and inability to infer causality. The study is observational and uncontrolled, making it impossible to distinguish the effect of SGLT2 inhibitors from the ongoing effects of ARNI, beta-blockers, or MRA therapy, or from the natural course of the disease. Without a comparator or historical data, any claims of “reverse remodeling” are unsupported and speculative. The conclusions attributing structural and functional improvement solely to SGLT2i use are therefore not justified.
A2: We thank the reviewer for this important comment. We fully acknowledge that the observational and uncontrolled design limits causal inference and that the absence of a comparator group precludes definitive attribution of the observed effects solely to SGLT2 inhibitor therapy. However, at the time of study initiation, SGLT2 inhibitors were already an integral component of guideline-directed medical therapy (GDMT) for HFrEF, supported by strong evidence from large, randomized trials and endorsed by professional societies. Therefore, establishing a control group not receiving SGLT2i would have been ethically unacceptable. Our study was designed to reflect real-world implementation of this therapy in a rare cardiomyopathy population (LVNC) under contemporary GDMT. This rationale has been clarified in the revised manuscript. (Page 3, Paragraph 3)
Q3:Insufficient data on prior medical therapy. The duration of treatment with ARNI, beta-blockers, and MRAs prior to study enrollment is not provided. The statement that patients received “maximally tolerated doses” lacks supporting data — actual dosages or dose ranges should be reported. The sentence “24 of patients were on angiotensin receptor/neprilysin inhibitors (ARNI), 1 on ACEI (due to ARNI adverse effect); all patients were receiving BB and MRA, 19 patients received dapagliflozin 10 mg, 6 patients empagliflozin 10 mg.” is misleading because implies prior use of these agents. This is inconsistent with the Methods section stating that SGLT2i were initiated at baseline.
A3: We thank the reviewer for this important observation. All patients were on stable, guideline-directed heart failure therapy—including ARNI, beta-blockers, and MRAs—for at least six months prior to initiation of SGLT2 inhibitor therapy. This stabilization period is consistent with clinical practice and supported by recent studies. The term “maximally tolerated doses” refers to individualized optimization according to patient tolerance and guideline recommendations. To enhance clarity, we have added a table specifying the actual dose ranges of each agent at baseline. We also acknowledge that the previous phrasing may have been misleading and have revised the manuscript to clearly state that SGLT2 inhibitors were initiated at baseline, while all other heart failure therapies were already ongoing and stable before study enrollment. This revision ensures full consistency between the Methods and Results sections. (Page 3, Paragraph 5 and Page 4, Paragraph 6)
Q4: Missing information on disease duration and clinical history. The manuscript provides no information on the time from heart failure diagnosis to SGLT2i initiation, which is essential for understanding the disease trajectory. Without these data, one cannot exclude the possibility that the observed improvements were due to prior optimization of conventional therapy rather than SGLT2 inhibition.
A4: We thank the reviewer for this relevant point. The interval between heart failure diagnosis and SGLT2i initiation varied across patients but was generally longer than one year. Conventional HFrEF therapy (ARNI, beta-blockers, and MRAs) had been optimized and stable for at least six months before SGLT2i introduction. This duration is typically sufficient for standard therapies to achieve their maximal clinical effect, supporting the interpretation that the observed improvements were attributable to the addition of SGLT2i rather than further optimization of background treatment. The manuscript has been revised accordingly. (Page 4, Paragraph 6)
Q5: Overstated and misleading conclusions. The authors conclude that SGLT2i therapy is associated with significant reverse remodeling and functional improvement. Based on the data presented, and considering the study’s uncontrolled design, small sample size, and major confounding factors, this statement is scientifically unjustified. There is strong evidence that SGLT2 inhibitors improve outcomes in patients with heart failure, so it could be expected that a similar effect would be observed in patients with LVNC (as the authors reported). However, in the population described here, there is no basis to attribute the observed effects solely to SGLT2 inhibitor therapy. The conclusions substantially overreach the presented evidence.
A5: We thank the reviewer for this comment and agree that our conclusions should be interpreted with caution. Given the observational and uncontrolled design, small sample size, and potential confounding factors, it is not possible to attribute the observed improvements solely to SGLT2 inhibitor therapy. Nevertheless, the direct effects of SGLT2 inhibitors on cardiomyocytes, together with their pleiotropic actions, likely contributed to the positive myocardial remodeling observed in our cohort. Appropriate revisions have been made in the manuscript. (Page 8, Paragraph 4; Page 9, Paragraph 7)
Round 2
Reviewer 1 Report
Comments and Suggestions for Authors
My congratulations to the efforts of the authors.
Reviewer 2 Report
Comments and Suggestions for Authors
Thanks for the responses and the manuscript improvement.
Reviewer 3 Report
Comments and Suggestions for Authors
The changes you have made have been found positive by me.